# Evaluation of Liquid Organic Acids on the Performance, Chyme pH, Nutrient Utilization, and Gut Microbiota in Broilers under High Stocking Density

**DOI:** 10.3390/ani13020257

**Published:** 2023-01-12

**Authors:** Miaomiao Han, Bingbo Chen, Yuanyang Dong, Zhiqiang Miao, Yuan Su, Ci Liu, Jianhui Li

**Affiliations:** 1College of Animal Science, Shanxi Agricultural University, Taigu 030801, China; 2College of Veterinary Medicine, Shanxi Agricultural University, Taigu 030801, China

**Keywords:** broilers, chyme pH, gut microbiota, nutrient utilization, organic acid, stocking density

## Abstract

**Simple Summary:**

Organic acids (OAs) are proposed as suitable alternatives in animal production. The released protons of OAs can reduce the pH in the gastrointestinal tract, then, nutrient digestion is stimulated by boosting the secretion of endogenous enzymes. In addition, lower intestinal pH suppresses pathogenic bacterial growth and encourages beneficial bacterial growth. In the starter phase, broilers are young, and their digestive tract is not well developed; thus, the first purpose of this study is to evaluate the efficacy of liquid OAs on the growth performance, serum metabolism lipid profiles, chyme pH, and duodenal enzymes activity in broilers during the starter phase. In addition, since a high stocking density (HSD) usually occurs in grower broilers and can lead to some disadvantages, whether supplemental OAs could alleviate the HSD stress condition in grower broilers is the second study aim of this study. Hence, the objective of the current study was to investigate whether the supplementation of liquid OAs in an entire period could alleviate the grower broilers’ stress status under an HSD. We found that OAs decreased the chyme pH in the upper digestive tracts and increased the activity of digestive enzymes in broilers during starter and grower phases. In addition, an HSD decreased the growth performance, while OAs showed no efficacy in the growth performance but improved the nutrient utilization in grower broilers. The alterations of gut microbiota were observed in grower broilers supplemented with OAs. In addition, the mechanisms of OAs in regulating the lipid metabolism and gut bacteria need further investigations.

**Abstract:**

This study aimed to evaluate the efficacy of organic acids (OAs) in starter broilers and to investigate whether supplemental OAs could alleviate the high stocking density (HSD) stress condition in grower broilers. A total of 408 1-day-old Arbor Acres broilers were assigned into two groups without or with liquid OAs in the starter phase. In the grower phase, each group in the starter phase was divided into a normal stocking density and an HSD. The OA dose was 0.16% at the starter and grower phases. The results showed that at the starter phase, OAs decreased the chyme pH in gizzard and duodenum and increased the activities of chymotrypsin and α-amylase in the duodenal chyme (*p* < 0.05). In the grower phase, an HSD decreased the growth performance and the ether extract digestibility (*p* < 0.01). The supplementation of OAs decreased the chyme pH in the gizzard, proventriculus, and duodenum and increased the lipase and α-amylase activities (*p* < 0.05). The supplemental OAs increased the dry matter and total phosphorous digestibility and the contents of acetic acids, butyric acids, isovaleric acids, and valeric acids (*p* < 0.05). For cecal microbial compositions at the genus level, an HSD decreased the relative abundance of *Blautia, Norank_f__norank_o__RF39,* and *Alistipes*, while supplemental OAs increased the relative abundance of *Norank_f__norank_o__RF39* (*p* < 0.05). In conclusion, although there were no interaction effects between OAs and stocking densities in the present study, it was clear that the supplementation of OAs has beneficial effects on the chyme pH, enzymes activities, and nutrient digestibility in broilers, while an HSD existed adverse effects on the growth performance, nutrient digestibility, and gut microbiota balance in grower broilers.

## 1. Introduction

With the increasing demand for poultry products, farmers pursue high yields and profitability by dramatically elevating the stocking densities in poultry production [1]. However, a high stocking density (HSD) leads to various negative effects in broilers [2]. The broilers’ behaviors of crouching, standing, and walking were limited under an HSD, which induces animal welfare problems [3]. In addition, our previous study illustrated that an HSD caused poor growth performances by decreasing the feed intake and daily weight gain, and this led to the occurrence of oxidative stress in the serum, liver, and jejunum of broilers [4]. Since an HSD is prone to oxidative injuries, a decreasing meat quality and growth performance occurred in broilers [5,6]. Moreover, an HSD could compromise the immune ability across different lymphoid organ disorders and decrease the humoral immune parameters [7,8], in addition to negatively affecting the intestinal barrier function and morphology [9,10]. Gut microbiota plays an integral role in the host’s health via its ability to modulate immune abilities or endocrine pathways [11,12]. It was illustrated that an HSD could affect the microbiota in broilers [5,13]. Therefore, the impaired intestinal physical function and disorder of microbiota in the intestine might lead to a lower nutrient absorption and digestibility.

Currently, antibiotics as growth promoters have been banned in the European Union and China in poultry production due to the antimicrobial resistance and pollution residues [14]. Organic acids (OAs) are proposed as the suitable alternatives for feed additives [15,16]. OAs are organic substances with a structure of carboxyl as weak acids [17]. The released proton of OAs can reduce the pH values in the gastrointestinal tract (GIT); then, nutrient digestion can be stimulated by boosting the secretion of endogenous enzymes [18]. In addition, lower pH values in the intestine suppress the growth of pathogenic bacteria and encourage beneficial bacterial growth [19]. Studies have illustrated that supplemental OAs can promote the growth performance and meat quality [20,21], improve nutrient utilization [22,23], and inhibit pathogenic bacteria [24,25]. The efficiency of OAs relies on the acid molecular, the dissociation constant, and the antimicrobial activity [26,27]. However, OAs added in diets have some disadvantages, such as corroding machines and volatilizing during diet granulation and storage [24,28]. Supplemental OAs in drinking water can reduce those problems and have multiple functions in water and animal health [24,28,29]. In addition, studies indicated that medium-chain fatty acids as a type of OA could improve early productive performances and animal welfare by reducing foot defects in Ross 308 broilers under an HSD (16 and 18 birds/m^2^) [30], but diets supplemented with 0.20% OAs (15% propionic acid, 24% formic acid, and 3% ammonium hydroxide) did not affect the heterophils, lymphocytes, and heterophils:lymphocytes ratio in brown laying hens under an HSD (287 cm^2^/hen) [31].

In the current study, the stocking density is defined by the number of broilers in a space and is associated with the age of the broilers [32]. There are generally no HSD stressful conditions in the early stage of broilers [2]. Therefore, the current experiment was designed according to the different stages of the broiler at the starter phase and grower phase. The objective of the present study was to evaluate the impact of an OA supplementation in drinking water on the performance, chyme pH, nutrient utilization, and microbiota of broilers under an HSD. Using an experimental design consisting of a 2 × 2 factorial arrangement with two levels of OAs (0 and 0.16%) provided during the grower phase along with two different stocking densities (14 birds/m^2^ and 20 birds/m^2^), we hypothesized that the supplementation of OAs in the entire period would improve the grower broilers’ health under an HSD that would be reflected in the positive effects observed in the performance, nutrient utilization, or gut microbiota.

## 2. Materials and Methods

### 2.1. Materials

The OAs (Selko-pH^®^) used in the current experiment were provided by Trouw Nutrition (Shanghai, China). Selko-pH^®^ is a pH-adjusting solution consisting of a proprietary mixture of formic acid (32%), acetic acid (7%), ammonium format (20%), mono- and diglyceride of unsaturated fatty acids, and copper acetate.

### 2.2. Animals and Experimental Design

The experiment was performed at the Daxiang Farming Group (Shanxi, China). A schematic drawing of the experimental design is shown in Figure 1. The experimental design consisted of a one-way analysis of variance of two groups at the starter phase and a 2 × 2 factorial design at the grower phase. Initially, a total of 408 one-day-old Arbor Acres broilers (Daxiang Farming Group Co., Shanxi, China) were randomly divided into 2 groups with 10 replicates in each group. In the control group (CON), broilers drank normal water without any OAs; in the organic acid group (OA), broilers drank the normal water supplemented with OAs. In the grower phase from 22 to 42 days of age, each group in the starter phase was randomly divided into two stocking densities (a normal stocking density (NSD, 14 birds/m^2^) and an HSD (20 birds/m^2^)). The 2 × 2 factorial management of the OAs and stocking density results in the following four treatment combinations: (1) CON + NSD: broilers drank the normal water under NSD; (2) CON + HSD: broilers drank the normal water under an HSD; (3) OA + NSD: broilers drank the OA-supplemented water under NSD; (4) and OA + HSD: broilers drank the OA-supplemented water under an HSD. Each group had six replicates at the grower phase. OAs was included in the drinking water in the starter and grower phases. The supplemental dose of Oas was determined by the pH titration in local, normal drinking water when the pH reached a value of 3.8 (about 0.16%). From day 1 to 42, OAs were not supplemented in drinking water on the day of the broilers’ immunization.

### 2.3. Management

All broilers were provided with commercial complete diets (Daxiang Farming Group Co., Shanxi, China) in crushed pelleted forms from day 1 to 22 and in pelleted forms from day 22 to 42. The broilers were raised in stainless steel cages with 70 cm× 70 cm in an environmentally controlled room. The room temperature was maintained at 34 °C for the first 3 days, after which the temperature was gradually reduced by 3 °C each week until it reached 21 °C; it was then maintained at this temperature until the end of the 42-day experiment. The relative humidity of the rearing room was kept between 50% and 70%. Twenty-three hours of light was maintained at first week, and then twenty hours of illumination was maintained per day for 2–4 weeks; finally, twenty-three hours of light per day until the end of the experiment was maintained. Feed and water were provided ad libitum throughout the study.

### 2.4. Growth Performance Measurement

On days 1, 21, and 42 of the experiment, the body weight (BW) and feed consumption of all chickens in each replicate were recorded after 12 h of fasting. The average daily gain (ADG), average daily feed intake (ADFI), and the ratio of feed to gain (FCR) were calculated on a replicate basis in the periods between day 1 and 21 and 22 and 42, respectively.

### 2.5. Sample Collection and Preparation

On day 21 and 42, one broiler that approximated the average BW in random six replicates was selected and euthanized by jugular vein bleeding after stunning using a 60% concentration of CO_2_ gas. The chyme of the gizzard, proventriculus, duodenum, jejunum, ileum, and cecum were collected to test the pH values. Moreover, the other chyme samples of the duodenum were harvested and stored at −80 °C to analyze the digestive enzyme activities.

From day 39 to 41, the excreta of each replicate were collected to analyze the nutrient digestibility using the endogenous indicator method. On day 42, blood was collected in 5 mL anticoagulant-free vacutainer tubes, and they were stood for 30 min at room temperature, then centrifuged at 3000× *g* for 15 min. Serum was collected and stored at −20 °C to analyze the serum lipid metabolism profiles. The cecal contents were gathered and immediately stored at −80 °C to analyze the volatile fatty acids (VFAs) contents and the high-throughput sequencing of microflora.

### 2.6. Determination of Serum Lipid Metabolism Profiles

On day 42, the serum contents of glucose (GLU), total triglyceride (TG), high-density lipoprotein cholesterol (HDLC), low-density lipoprotein cholesterol (LDLC), and cholesterol (TC) were analyzed using an automatic biochemical analyzer (Hitachi 7160, Hitachi High-Technologies Corporation, Tokyo, Japan).

### 2.7. Determination of Chyme pH and Enzyme Activity

The pH values in the chyme from the gizzard, proventriculus, duodenum, jejunum, ileum, and cecum were analyzed by using a pH meter (testo201, Germany) based on the method reported by Wu et al. [33].

The chyme from the duodenum was precisely weighted and diluted to a 1 to 9 weight/volume using a 0.9% normal saline solution. Afterwards, the mixture was homogenized in ice water and centrifuged at 4 °C for 10 min at 3000 rpm/min; then, the supernatant was collected to detect the enzyme activity in the duodenal chyme. The activities of trypsin (Kit No. A080-2), chymotrypsin (Kit No. A080-3), lipase (Kit No. A054-1-1), and amylase (Kit No. C016-1-1) in the duodenum were detected with the corresponding commercial kits (Nanjing Jiancheng Bioengineering Institute, Nanjing, China) according to the manufacturer’s instructions.

### 2.8. Nutrient Digestibility

The nutrient digestibility was analyzed by using an endogenous indicator of the hydrochloric acid insoluble ash (AIA) method reported by Sales and Janssens [34]. The excreta collected from day 39 to 41 was dried at 60 °C–65 °C, ground through a 0.5 mm screen by a mill grinder, and stored at −20 °C for further analyses. The contents of the moisture (method 930.15), crude protein (990.03), ether extract (method 920.39), calcium (method 968.08), and phosphorous (method 965.17) in the diet and excreta were detected according to the procedures based on the Association of Official Analytical Chemists [35]. The hydrochloric acid insoluble ash contents in the diet and excreta were analyzed according to the method reported by Zhang et al. [36]. The digestibility of each nutrient was calculated as described by Frame et al. [37] according to the following formula.
$$\mathrm{Digestibility}\left(\%\right)=\left(1-\frac{{\mathrm{marker}}_{\mathrm{feed}\;\left(\%\right)}\times {\mathrm{nutrient}}_{\mathrm{excreta}\;\left(\%\right)}}{{\mathrm{marker}}_{\mathrm{excreta}\;\left(\%\right)}\times {\;\mathrm{nutrient}}_{\mathrm{feed}\;\left(\%\right)}}\right)\times 100$$

### 2.9. Determination the VFAs Contents

The VFAs contents in the cecum were detected by gas chromatography according to the method reported by Wang et al. [38]. The internal standard solution of 2-ethylbutyric acid and the external standard solution containing six acids (acetic acid, propionic acid, isobutyric acid, butyric acid, isovaleric acid, and valeric acid), and these were purchased from Sigma-Aldrich (Shanghai, China). Approximately 1 g of thawed cecal chyme was weighed and placed into a 10 mL polypropylene tube and supplemented with 1% hydrochloric acid and 5% formic acid. The solution was homogenized and placed in an ice bath for 30 min to settle the protein completely and then centrifuged at 1500 rpm for 10 min. The solution was transferred to a 1.5 mL centrifuge tube to undergo centrifugation again at 14,000 rpm for about 10 min. The supernatant was prepared to detect the VFAs contents using an internal standard of 2-ethylbutyric acid by gas chromatography (6890 Series GC System, HP, Palo Alto, CA, USA) equipped with a 30 m × 0.25 mm × 0.25 µm column (DB-FFAP, Agilent Technologies, Inc, Santa Clara, CA, USA) and a flame ionization detector after being filtered through a 0.22 µm filter.

### 2.10. Cecal Microbiota Analyses

The total microbial genomic DNA was extracted from cecal chyme samples using the E.Z.N.A.^®^ soil DNA Kit (Omega Bio-tek, Norcross, GA, USA) according to the manufacturer’s instructions. The quality and concentration of the DNA were determined by 1.0% agarose gel electrophoresis and a NanoDrop^®^ ND-2000 spectrophotometer (Thermo Scientific Inc., USA) and kept at −80 ℃ prior to further use. The hypervariable region V3–V4 of the bacterial 16S rRNA gene was amplified with primer pairs 338F (5’-ACTCCTACGGGAGGCAGCAG-3’) and 806R(5’-GGACTACHVGGGTWTCTAAT-3’) by an ABI GeneAmp^®^ 9700 PCR thermocycler (ABI, CA, USA). Purified amplicons were pooled in equimolar amounts and paired-end sequenced on an Illumina MiSeq PE300 platform/NovaSeq PE250 platform (Illumina, San Diego, CA, USA) according to the standard protocols by Majorbio Bio-Pharm Technology Co. Ltd. (Shanghai, China). Raw FASTQ files were demultiplexed using an in-house perl script, and then quality-filtered by fastp version 0.19.6 and merged by FLASH version 1.2.7. The optimized sequences were clustered into operational taxonomic units (OTUs) using UPARSE 7.1 with a 97% sequence similarity level. The most abundant sequence for each OTU was selected as a representative sequence. The bioinformatic analyses of the gut microbiota were carried out using the Majorbio Cloud platform (https://cloud.majorbio.com (accessed on 9 November 2020)).

### 2.11. Statistical Analysis

A statistical analysis was performed using SAS statistics (version 9.4, SAS Inst. Inc., Cary, NC, USA). In the starter phase, an independent sample *t*-test (Student’s *t* test) was used for comparing the two groups. The normality of the data was assessed using the Shapiro–Wilk test (W > 0.05). Each replicate served as the experimental unit. In the grower phase, a two-way ANOVA was used to evaluate the main effects (OAs and stocking density). The data were analyzed by MIXED procedures. Tukey’s post hoc test was applied to compare the treatment’s means. In starter and grower phases, differences were considered significant at the level of *p* ≤ 0.05, whereas 0.05 < *p* < 0.10 was considered as a tendency. In addition, a correlation among cecal microbiota and the parameters of growth performance, lipid metabolism profiles, pH values in chyme, enzyme activities, and VFA contents was considered to be statistically robust if Spearman’s correlation coefficient was over 0.6 or less than −0.6 and if the *p* -value was less than 0.01.

## 3. Results

### 3.1. Growth Performance

Data on the growth performance are listed in Table 1. In the starter phase, OAs had no significant effects on the BW, ADG, and ADFI of broilers in the starter phase but tended to decrease the FCR compared with the CON group (*p* = 0.059). The high-density condition induced the poor performance by decreasing the 42 d BW (*p* = 0.003), ADG (*p* = 0.002), and ADFI (*p* < 0.001) in the grower phase. An OA supplementation in drinking water does not exert effects on the performance in broilers.

### 3.2. Serum Lipid Metabolism Profiles

As illustrated in Table 2, the supplementation of OAs significantly reduced the serum TG (*p* = 0.028) and LDLC contents (*p* = 0.050) in broilers at the starter phase. In the grower phase, the HSD increased the serum LDLC concentrations in broilers (*p* = 0.050). There are interaction effects in serum TG and HDLC. The OA + HSD group had much higher serum TG contents than the other three groups (*p* = 0.037). Compared with the CON + NSD group, the serum HDLC contents significantly decreased in the OA + NSD group, but there were no observed differences in the CON + NSD, CON + HSD, and OA + HSD groups (*p* = 0.048).

### 3.3. Chyme pH Values

As illustrated in Table 3, an OA supplementation significantly reduced the chyme pH value in the gizzard (*p* = 0.022) and duodenum (*p* < 0.001) of the broilers in the starter phase. The chyme pH value in proventriculus tended to decrease in the OA group (*p* = 0.095). In the grower phase, an HSD did not affect the chyme pH values in different digestive tract parts. However, an OA supplementation decreased the chyme pH value in the gizzard, proventriculus, and duodenum (*p* < 0.05) in the broilers. There were no interaction effects between the OA and stocking density in the chyme pH value at the grower phase.

### 3.4. Enzyme Activities in Duodenum

The effects of organic acids on the enzyme activities in the duodenal chyme in the starter and grower broilers under an HSD are listed in Table 4. In the starter phase, the activities of chymotrypsin (*p* = 0.043) and α-amylase (*p* = 0.046) were significantly elevated in the OA group compared with the CON group. The trypsin and lipase activities were not affected in the broilers. In the grower phase, the supplementation of OAs in drinking water significantly increased the lipase (*p* = 0.028) and α-amylase (*p* = 0.033) activities in duodenal chyme. There were no interaction effects between the organic acids and stocking density in the duodenal enzyme activities.

### 3.5. Nutrient Digestibility and VFAs Contents

As shown in Table 5, the HSD significantly decreased the EEs (*p* = 0.012) digestibility and tended to decrease the CPs digestibility (*p* = 0.057) in the grower broilers. The supplementation of OAs significantly elevated the DMs (*p* = 0.047) and TPs (*p* = 0.042) digestibility. In addition, the digestibility of the Ca (*p* = 0.056) increased at a trend by supplementing the OAs in water. There were no interaction effects between the OAs and the stocking density in the nutrient digestibility during the grower phase.

As illustrated in Table 6, there were no main effects of the stocking density and interactions on the VFAs contents in cecal chyme. Supplemental OAs in drinking water increased the levels of acetic acid (*p* = 0.019), butyric acid (*p* = 0.001), isovaleric acid (*p* = 0.001), and valeric acid (*p* = 0.003) in cecum.

### 3.6. Microbial Diversity and Community in Cecal Chyme

Following quality control, a total of 1,428,036 reads were filtered using QIIME. The rarefaction curves tended to approach the asymptote, suggesting a sufficient depth for sequencing saturation (Figure 2a). The supplementation of OAs and the stocking density did not affect the alpha diversity of the cecal microbiota of broilers (Appendix A). The beta diversity analysis performed via principal coordinates analysis (PCoA) could not be differentiated from the microbial community among the groups (Figure 2b). At the phylum level (relative abundance > 0.5%), Firmicutes (69.46–81.36%), Bacteroidota (17.12–28.79%), Cyanobacteria (0.39–0.59%), and Actinobacteriota (0.39–0.65%) accounted for 98% of the total cecal microbiota (Figure 3a and Appendix A). At the genus level (Figure 3b and Appendix A), the high-density challenge decreased the relative abundance of *Blautia* (*p* = 0.004), *Norank_f__norank_o__RF39* (*p* = 0.040), and *Alistipes* (*p* = 0.038). Supplemental OAs in drinking water increased the relative abundance of *Norank_f__norank_o__RF39* (*p* = 0.011) and tended to increase the relative abundance of *Christensenellaceae_R-7_group* (*p* = 0.051), *Norank_f__norank_o__Clostridia_UCG-014* (*p* = 0.091), and *UCG-005* (*p* = 0.061). There are interaction effects in the relative abundance of *Alistipes* (*p* = 0.031). The relative abundance of *Alistipes* was much lower in the CON + HSD than in the CON + HSD group, while no differences exist with the OA + NSD and OA + HSD groups.

### 3.7. Association between Cecal Microbiota, and the Parameters Tested on Day 42

A heatmap for Spearman’s correlation was produced to determine if any relationship exists among cecal microbiota and the parameters of the growth performance, serum lipid metabolism profiles, pH value in chyme, nutrient digestibility, enzyme activities, and VFA contents during the grower phase (Figure 4). For the growth performance, *Barnesiella* was positively associated with 42 d BW, while FCR was negatively associated with *UCG-005* (*p* < 0.05). For the serum lipid metabolism profiles, the levels of LDLC and TC were positively associated with the *norank_f__Eubacterium_coprostanoligenes_group.* The level of TG was positively associated with *Subdoligranulum* (*p* < 0.05). For chyme pH, the pH value in the gizzard was positively associated with *Barnesiella* and negatively associated with *norank_f__norank_o__RF39*. The pH value in proventriculus was positively associated with *Streptococcus*. The pH value in the jejunum was negatively associated with *Rikenella* and *norank_f__norank_o__Clostridia_vadinBB60_group* (*p* < 0.01). The pH value in the ileum was negatively associated with *Rikenella* and *Subdoligranulum*. For the nutrient digestibility, the digestibility of DM and TP was positively associated with *Subdoligranulum*. The digestibility of EE was negatively associated with *norank_f__Ruminococcaceae* and *Bacteroides*. The activities of α-amylase and lipase were positively associated with *Christensenellaceae_R-7_group* and negatively associated with *Enterococcus* (*p* < 0.05). For the VFAs, the level of butyric acid was positively associated with *Alistipe* (*p* < 0.05).

## 4. Discussion

High stocking densities are common stress conditions observed in poultry production. Studies have confirmed that an HSD has negative effects on the growth performance, animal behaviors, nutrient digestibility, and gut microbiota in broilers [2,3,4,5,6,7,8]. Therefore, nutrition strategies are important to prevent stress injuries in broilers under an HSD. Organic acids have the characteristics of being pollution-free, drug resistance-free, and residual-free and have beneficial effects on the health of broilers [16,18]. Thus, we analyzed the effects of OAs in the drinking water of broilers at the starter phase and grower phase with an HSD. In the present study, supplemental OAs in drinking water showed no growth-promoting effects in the starter phase, but the FCR in the OA group tended to decrease by 2.13%. Since an HSD easily occurred in older broilers [2], we added HSD stress in the grower phase to evaluate the efficacy of the OAs in the drinking water.

At present, the growth performances were reduced in broilers at the grower phase under an HSD by decreasing 42 d BW, ADG, and ADFI, but supplemental OAs did not exert any significant improvement in the growth performance. Therefore, the current study illustrated that an HSD led to the resulting stress status of the broilers, but the OAs inclusion in drinking water did not affect the growth performance in the starter and grower phases, which was consistent with the results reporting that the drinking water’s acidification did not have any beneficial effects on the performances in Ross 308 broilers under heat stress [39]. Studies about supplemental OAs in the drinking water of broilers under different conditions are not similar with one another. The inclusion of 0.1% OAs (formic acid, acetic acid, and ammonium formate) in drinking water elevated the growth performances in Arbor Acres broilers [40], while supplemental 0.1%, 0.15%, and 2% OAs (ortho phosphoric acid, formic acid, and propionic acid) in the drinking water decreased the growth performance in broilers [41]. There exist no growth-promoting effects in the broilers supplemented with 0.075% OAs (formic acid, acetic acid, propionic acid, and sorbic acid; medium-chain fatty acids combined with ammonium formate; and coconut/palm kernel fatty acid distillate) in their water [42]. Bourassa et al. [43] reported that formic acid in water (1%) decreased the body weight in Cobb 500 broilers challenged with *Salmonella Typhimurium*, whereas supplemental Selko-pH in drinking water (1%) increased the feed intakes during day 1–49 but did not affect the weight gain and feed conversion ratio in Cobb 500 broilers that were orally challenged with (10^9^ cfu/mL) *Campylobacter jejuni* [44]. In addition, several studies showed that supplemental OAs in water can improve the growth performances in broilers challenged with *coccidium* [29] and *Salmonella Pullorum* [45], and broilers who were fed diets with different levels of threonine [46] and different wheat grain forms [47]. These inconsistent results may be attributed to the local water quality, the levels of the supplemental OAs, broiler breeds, and feed nutrition levels.

In the starter phase, supplemental OAs in drinking water significantly decreased the serum TG and LDLC contents compared to the CON group, which indicated that OAs could effectively regulate the lipid metabolism. However, in the grower phase, the inclusion of OAs increased the serum TG contents, which is consistent with the results reported by Baltić et al. [48], who illustrated that supplemental medium-chain fatty acids increased the TG contents in Cobb 500. In addition, the inclusion of OAs also increased the serum GLU contents, which is consistent with Iqbal et al. [49], who observed that a diet supplemented with 1 g/kg of OAs (formic acid 40%, formate 40%, and sodium 20%) increased the serum GLU contents in Ross 308 broilers. A possible justification could be that OAs could motivate the insulin secretion and stimulated anabolic-related processes [50]. However, the discrepancies of the OAs in the serum TG contents in the starter phase and grower phase were unclear and the mechanism requires further investigations regarding the lipid metabolism.

As is well known, OAs as weak acids can release protons to decrease the pH values in GIT [18]. In the current study, supplemental OAs in drinking water decreased the chyme pH value in the gizzard and duodenum. The chyme pH value in proventriculus tended to decrease in the OA group of broilers in the starter phase. In addition, OAs reduced the chyme pH value in the gizzard, proventriculus, and duodenum in grower broilers. Thus, supplemental OAs in drinking water can reduce the chyme pH values in the upper digestive tract but not in the entire digestive tract, which might be a result of the observation that the upper segments of the GIT can metabolize and rapidly absorb the OAs [51]. Thereafter, OAs could not present efficacies in the latter GIT. Several studies also indicated that diet or water supplemented with OAs could decrease the broiler digestive tract’s pH [52,53]. Since young broilers have a limited ability in producing enough acid to maintain the optimal pH values in the GIT to activate enzymes, ensure nutrition digestion, and maintain the balance of the gut microbiota in the intestine [54], supplemental OAs can activate the enzyme precursors in the acid environment. Thus, the activities of chymotrypsin and α-amylase were elevated in the broilers during the starter phase in the current study. At the grower phase, an HSD did not cause any negative effects on the enzyme activities in broilers, while supplemental OAs increased the activities of lipase and α-amylase in the duodenal chyme, which might be due to the obvious decrease in the chyme’s pH value in the duodenum in this study. The similar results of OAs in feeds promoting digestive enzyme activities were also observed by Adil et al. [55]. However, there were no effects on the digestive enzyme activities by a dietary supplementation with 0.2% OAs (32% fumaric acid, 3% formic acid, 13% lactic acid, 3% propionic acid, and 1% citric acid) in Ross 308 broilers [56]. The discrepancies in those studies might be due to the component of OAs and the acid dissociation constant.

The current study demonstrated that an HSD decreased the EE digestibility in grower broilers; however, as aforementioned, the OAs in water decreased the chyme pH values and increased the digestive enzymes activities. Thus, the increasing nutrient digestibility of DM and TP were observed in the OA group in grower broilers, and the Ca digestibility also tended to increase in grower broilers fed with OAs in drinking water. Therefore, supplemental OAs in drinking water could elevate the nutrient utilization in grower broilers. It was reported that the diet inclusion of 1–3 g/kg (0.1–0.3%) of OAs (formic acid, lactic acid, malic acid, tartaric acid, citric acid, and orthophosphoric acid) improved the apparent metabolizable energies and total phosphorous ileal digestibility in Ross broilers [57]. Several studies also demonstrated that OAs can improve the nutrient utilization in broilers [14,22,23]. OAs can release protons and reduce the pH in GIT and then improve the digestive enzymes activities to elevate the nutrient digestibility. In addition, lower pH values can also reduce pathogenic bacteria, ultimately leading to improvements in the nutrient digestibility [58].

It is well known that carbohydrates in feeds cannot be entirely absorbed by animals, and undigested carbohydrates can be fermented to produce the short-chain fatty acids in the intestine tract by microbiota [59,60]. In turn, short-chain fatty acids can present decisive effects when regulating the microbial population in the GIT, elevating the immune system and promoting a resistance to inflammation [61]. Valeric acid and butyric acid, as short-chain fatty acids, have abilities against Gram-negative or Gram-positive bacteria [62]. In addition, butyrate can regulate the gut barrier and play anti-inflammatory and immunoregulatory roles to maintain gut homeostasis [62]. Ma et al. [60] indicated that a diet supplemented with 0.3% OAs (11% formic acid, 13% ammonium formate, 5.1% acetic acid, 10% propionic acid, 4.2% lactic acid, and ≤ 2% of other lower levels of organic acids (sorbic acid and citric acid)) increased the formic acid in cecal contents on day 21 and acetic acid, propionic acid, butyric acid, and the total volatile fatty acids in the cecal content on day 42. Our results illustrated that supplemental OAs in drinking water elevated acetic acid, butyric acid, and isovaleric acid but decreased valeric acid concentrations of the cecal chyme in broilers during the grower phase, which is probably correlated with changes in the gut microbiota in broilers supplemented with OAs in water.

The gut microflora plays vital roles in the health of animals by regulating nutrient utilization, intestinal integrity, and immune functions [24]. On the one hand, OAs can control pathogens by reducing the GIT pH; on the other hand, OAs could release proton irons in the cytoplasm. Then, pH-sensitive bacteria are forced to discard redundant proton irons via the H^+^-ATPase pump, and the bacteria proliferation is impeded [18]. The used OAs in this study mainly comprise formic acid (32%), acetic acid (7%), and ammonium formate (20%). It was illustrated that formic acid could show beneficial effects on the physical growth, digestibility, immunity, and antimicrobial activity, and acetic acid has positive effects on the anti-bacteria for animals [58]. Formic acid and acetic acid can directly control pathogens by acting upon the cell wall of Gram-negative bacteria [18]. It was indicated that supplemental Selko-pH in drinking water (0.1%) decreased the *Campylobacter* counts in the d 49 cecal content and d 35 and d 42 fecal contents in Cobb 500 broilers that were orally challenged with (10^9^ cfu/mL) *Campylobacter jejuni* on day 21 [38]. In addition, a dietary supplementation of 0.2% OAs (32% fumaric acid, 3% formic acid, 13% lactic acid, 3% propionic acid, and 1% citric acid) decreased the *Escherichia coli* population and increased the *Lactobacillus spp.: Escherichia coli* ratio in the ileum and caecum of Ross 308 broilers [50]. It was indicated that Gram-negative bacteria, such as *Campylobacter* and *Escherichia coli*, are susceptible to OAs with fewer than eight carbons [63]. Therefore, OAs have the ability to maintain the gut’s health. Hu et al. [40] indicated that the relative abundance of the Bacteroidetes, Firmicutes, and the Firmicutes/Bacteroidetes ratio was not affected by supplemental OAs (formic acid, acetic acid, and ammonium formate) in water and in diets, but the relative abundance of *Proteobacteria, Verrucomicrobia*, and *Cyanobacteria* decreased and actinobacteria increased in broilers that were supplemented with OAs in water. However, in the present study, the cecal microbiota’s composition at the phylum level was not affected by supplementing OAs in the water of grower broilers under an HSD. Multiple factors, such as different types of OAs or blends, may contribute to the divergences in the composition and relative abundance for gut microbiota [40]. At the genus level, *Blautia* plays an important role in alleviating the host’s disease via mitigating inflammation [64]. High-density challenges decreased the relative abundance of *Blautia*, which is consistent with the results reported by Wu et al. [5]. In addition, an enrichment of butyrate-producing bacteria, *Alistipes*, displayed a favorable anti-inflammatory effectiveness in human and laboratory animals [65,66]. The *Alistipes* relative abundance significantly decreased in the HSD group with normal water, which indicated that high stocking densities might influence energy utilizations. Thus, an HSD decreased the cecal beneficial bacteria in broilers. In addition, the OA supplement in drinking water increased the relative abundance of *Norank_f__norank_o__RF39* and tended to elevate the relative abundance of *Christensenellaceae_R-7_group*, *Norank_f__norank_o__Clostridia_UCG-014*, and *UCG-005*. Spearman’s correlation reflected that the activities of α-amylase and lipase were positively associated with the *Christensenellaceae_R-7_group*, while FCR was negatively associated with *UCG-005*. Moreover, altered genera with increasing relative abundances might be potentially related with body fat and the energy metabolism [67]. In addition, it was speculated that the *Christensenellaceae R-7 group* may be related to the glucose and amino acid metabolism [68]. Therefore, OAs could improve the nutrient digestibility by improving the enzyme activities and altering some metabolism-related bacteria in the current study.

## 5. Conclusions

In conclusion, the supplementation of OAs in drinking water did not affect the growth performances but decreased the chyme pH in the upper digestive tract and increased the enzyme activities in the duodenal chyme in broilers during the starter and grower phases. In addition, an HSD showed negative effects on the growth performances and nutrient digestibility and decreased the cecal beneficial bacteria in the broilers. The supplementation of OAs in drinking water improved the nutrient digestibility in the grower broilers. In addition, further research may need to choose a higher dose of OAs in water than the level used in the current study; exploring the fundamental mechanism for alleviating the stressful condition of OAs in grower broilers under an HSD is also suggested, such as the microbiota components in all digestive tracts.

## Figures and Tables

**Figure 1 animals-13-00257-f001:**
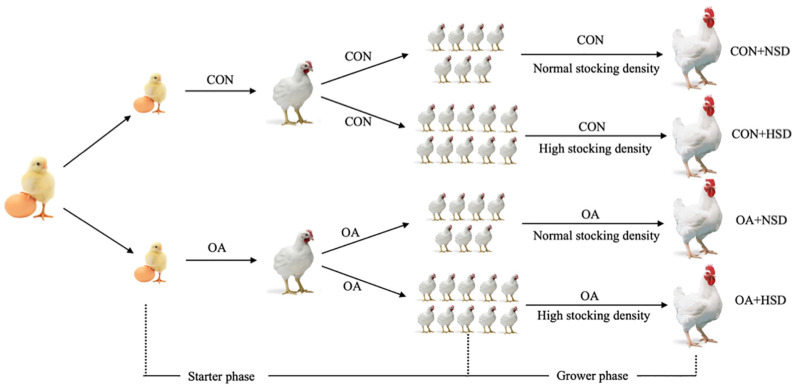
Schematic drawing of the experimental design. In the starter phase, broilers were randomly divided into two groups: broilers that drank normal water without any organic acids (CON) and organic acids in drinking water (OA). In the grower phase, each group in the starter phase was randomly divided at two stocking densities. The factorial management of organic acids and stocking density results in the following four treatment combinations: (1) CON + NSD: broilers that drank normal water under normal stocking density; (2) CON + HSD: broilers that drank normal water under high stocking density; (3) OA + NSD: broilers that drank OA-supplemented water under normal stocking density; (4) OA + HSD: broilers that drank OA-supplemented water under high stocking density.

**Figure 2 animals-13-00257-f002:**
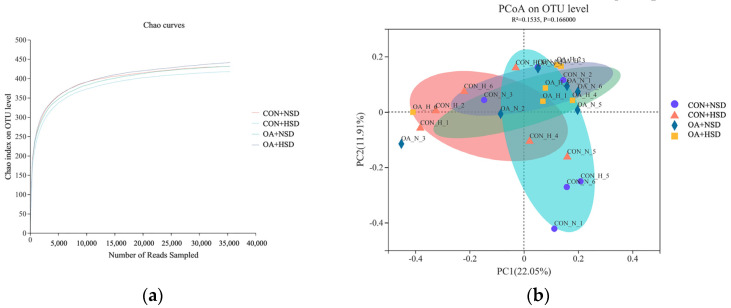
Rarefaction curves (**a**) and effects of organic acids on the beta diversity of cecal bacteria in broilers under high stocking density in the grower phase (**b**). CON + NSD, broilers drank normal water under normal stocking density; CON + HSD, broilers drank normal water under high stocking density; OA + NSD, broilers drank organic acid-supplemented water under normal stocking density; OA + HSD, broilers drank organic acid-supplemented water under high stocking density.

**Figure 3 animals-13-00257-f003:**
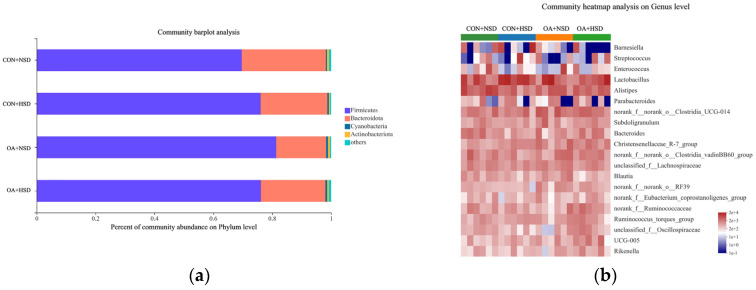
Effects of organic acids on the distribution of cecal bacteria at the phylum (**a**) and genus (**b**) levels in broilers under high stocking density in the grower phase. CON + NSD, broilers drank normal water under normal stocking density; CON + HSD, broilers drank normal water under high stocking density; OA + NSD, broilers drank organic acid-supplemented water under normal stocking density; OA + HSD, broilers drank organic acid-supplemented water under high stocking density.

**Figure 4 animals-13-00257-f004:**
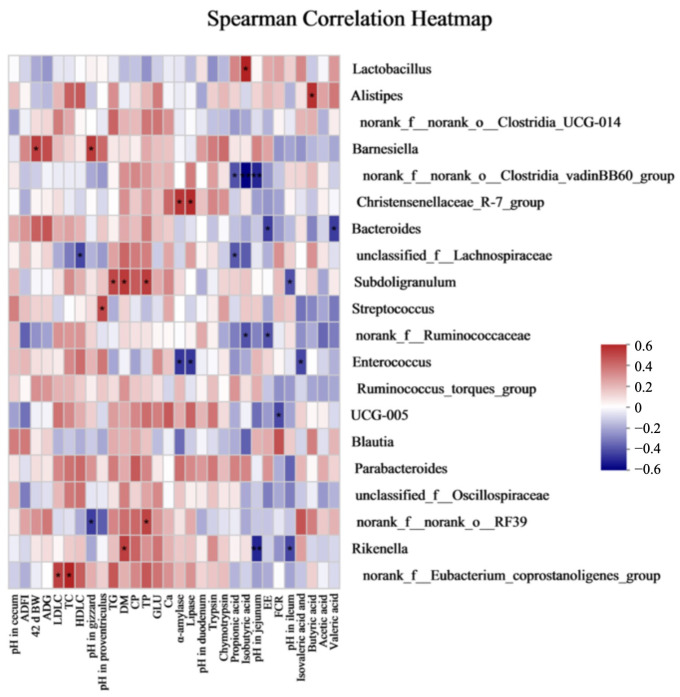
Heatmap of Spearman’s correlation among cecal microbiota and the parameters of growth performance, lipid metabolism profiles, pH value in chyme, enzyme activities, and VFA contents during the grower phase. Red suggests a positive correlation, and green suggests a negative correlation. “*” indicates 0.01 < *p* ≤ 0.05 and “**” indicates 0.001 < *p* ≤ 0.01.

**Table 1 animals-13-00257-t001:** Effects of organic acids on growth performance in broilers at starter and grower phases ^1^.

Starter Phase	CON		OA		SEM	*p*-Value		
0 d BW, g	46.84		46.02		0.545	0.496		
21 d BW, g	616.00		626.50		6.098	0.409		
ADG, g	27.10		27.64		0.284	0.362		
ADFI, g	38.31		37.96		0.222	0.443		
FCR	1.41		1.38		0.010	0.059		
**Grower phase**	**CON + NSD**	**CON + HSD**	**OA + NSD**	**OA + HSD**	**SEM**	**OA**	**Density**	**INT**
42 d BW, g	2259.71	2081.20	2277.34	2163.66	25.961	0.259	0.003	0.461
ADG, g	81.44	72.66	81.28	75.87	1.193	0.443	0.002	0.396
ADFI, g	117.86	109.82	123.42	109.62	1.677	0.294	<0.001	0.260
FCR	1.45	1.51	1.52	1.45	0.015	0.989	0.861	0.139

^1^ Data represent the means of ten replicates (*n* = 10) at starter phase and six replicate (*n* = 6) at grower phase. CON, control group; OA, organic acids group; NSD, normal stocking density; HSD, high stocking density; BW, body weight; ADG, average daily gain; ADFI, average daily feed intake; FCR, the ratio of feed to gain; SEM, standard error of mean. INT, interaction.

**Table 2 animals-13-00257-t002:** Effects of organic acids on serum lipid metabolism profiles in broilers at starter and grower phases ^1^.

Starter Phase	CON		OA		SEM	*p*-Value		
GLU, mmol/L	10.81		10.00		0.933	0.692		
TC, mmol/L	3.60		3.99		0.153	0.224		
TG, mmol/L	1.37 *		0.79		0.139	0.028		
HDLC, mmol/L	2.65		2.79		0.079	0.393		
LDLC, mmol/L	0.68 *		0.45		0.059	0.050		
**Grower phase**	**CON + NSD**	**CON + HSD**	**OA + NSD**	**OA + HSD**	**SEM**	**OA**	**Density**	**INT**
GLU, mmol/L	9.47	8.76	10.29	11.02	0.307	0.009	0.987	0.196
TC, mmol/L	3.01	3.20	2.64	3.06	0.092	0.161	0.095	0.505
TG, mmol/L	0.41 ^b^	0.44 ^b^	0.67 ^a^	0.47 ^b^	0.032	0.009	0.093	0.037
HDLC, mmol/L	2.39 ^a^	2.35 ^a^	1.94 ^b^	2.33 ^ab^	0.060	0.030	0.098	0.048
LDLC, mmol/L	0.53	0.71	0.52	0.62	0.035	0.455	0.050	0.547

^1^ Data represent the means of six replicates (*n* = 6). * Means *t*-test results of independent samples in CON and OA groups at starter phase (* *p* < 0.05). ^a, b^ Means in the same row without the same superscripts differ significantly (*p* < 0.05). CON, control group; OA, organic acids group; NSD, normal stocking density; HSD, high stocking density; SEM, standard error of mean. INT, interaction. GLU, glucose; TC, total cholesterol; TG, total triglyceride; HDLC, high-density lipoprotein cholesterol; LDLC, low-density lipoprotein cholesterol.

**Table 3 animals-13-00257-t003:** Effects of organic acids on the chyme pH values in broilers at starter and grower phases ^1^.

Starter Phase	CON		OA		SEM	*p*-Value		
Gizzard	3.76 *		3.59		0.039	0.022		
Proventriculus	3.86		3.75		0.032	0.095		
Duodenum	5.86 **		5.81		0.008	<0.001		
Jejunum	6.17		6.16		0.007	0.528		
Ileum	6.46		6.45		0.006	0.683		
Cecum	6.81		6.81		0.007	0.936		
**Grower phase**	**CON + NSD**	**CON + HSD**	**OA + NSD**	**OA + HSD**	**SEM**	**OA**	**Density**	**INT**
Gizzard	3.64	3.64	3.46	3.50	0.024	<0.001	0.634	0.602
Proventriculus	3.82	3.82	3.74	3.74	0.012	<0.001	0.798	0.865
Duodenum	5.90	5.91	5.83	5.80	0.015	0.003	0.706	0.572
Jejunum	6.22	6.22	6.21	6.20	0.008	0.356	0.780	0.534
Ileum	6.54	6.62	6.57	6.55	0.018	0.516	0.443	0.181
Cecum	6.75	6.76	6.70	6.68	0.018	0.067	0.944	0.656

^1^ Data represent the means of six replicates (*n* = 6). * Means *t*-test results of independent samples in CON and OA groups at the starter phase (* *p* < 0.05, ** *p* < 0.01). CON, control group; OA, organic acids group; NSD, normal stocking density; HSD, high stocking density; SEM, standard error of mean; INT, interaction.

**Table 4 animals-13-00257-t004:** Effects of organic acids on enzyme activities in duodenal chyme in broilers at the starter and grower phases ^1^.

Starter Phase	CON		OA		SEM	*p*-Value		
Trypsin, U/mg	1945.37		2307.80		235.281	0.468		
Chymotrypsin, U/mg	2.40		3.64 *		0.314	0.043		
Lipase, U/g	31.16		39.74		6.611	0.542		
α-amylase, U/mg	0.025		0.039 *		0.004	0.046		
**Grower phase**	**CON + NSD**	**CON + HSD**	**OA + NSD**	**OA + HSD**	**SEM**	**OA**	**Density**	**INT**
Trypsin, U/mg	2406.24	3061.68	5176.49	2932.19	422.608	0.103	0.316	0.075
Chymotrypsin, U/mg	2.39	2.72	3.27	2.26	0.331	0.766	0.621	0.340
Lipase, U/g	34.57	40.60	282.09	103.49	36.956	0.028	0.202	0.174
α-amylase, U/mg	0.02	0.02	0.03	0.03	0.003	0.033	0.897	0.826

^1^ Data represent the means of six replicates (*n* = 6). * Means *t*-test results of independent samples in CON and OA groups at starter phase (* *p* < 0.05). CON, control group; OA, organic acids group; NSD, normal stocking density; HSD, high stocking density; SEM, standard error of mean; INT, interaction.

**Table 5 animals-13-00257-t005:** Effects of organic acids on nutrient digestibility in broilers under high stocking density in the grower phase (%) ^1^.

Items	CON + NSD	CON + HSD	OA + NSD	OA + HSD	SEM	*p*-Value		
						OA	Density	INT
DM	66.31	66.06	70.02	67.14	0.621	0.047	0.181	0.257
EE	85.72	82.03	86.08	84.25	0.569	0.153	0.012	0.355
CP	53.57	52.48	59.09	53.07	0.997	0.099	0.057	0.176
TP	30.38	24.93	35.88	32.95	1.665	0.042	0.190	0.688
Ca	19.51	25.55	30.34	29.08	1.857	0.056	0.505	0.313

^1^ Data represent the means of six replicates (*n* = 6). CON, control group; OA, organic acids group; NSD, normal stocking density; HSD, high stocking density; SEM, standard error of the means; INT, interaction; DM, dry matter; EE, ether extract; CP, crude protein; TP, total phosphorous; Ca, calcium.

**Table 6 animals-13-00257-t006:** Effects of organic acids on volatile fatty acids contents in broilers under high stocking density in grower phase (mmol/L) ^1^.

Items	CON + NSD	CON + HSD	OA + NSD	OA + HSD	SEM	*p*-Value		
						OA	Density	INT
Acetic acid	3.55	2.14	3.96	3.74	0.236	0.019	0.052	0.147
Propionic acid	0.81	0.69	0.77	0.81	0.040	0.653	0.629	0.340
Isobutyric acid	0.14	0.15	0.14	0.15	0.007	0.944	0.466	0.949
Butyric acid	1.13	0.94	1.79	1.33	0.106	0.008	0.085	0.451
Isovaleric acid	0.16	0.16	0.20	0.19	0.005	0.001	0.635	0.494
Valeric acid	0.16	0.14	0.22	0.20	0.010	0.003	0.289	0.963

^1^ Data represent the means of six replicates (*n* = 6). CON, control group; OA, organic acids group; NSD, normal stocking density; HSD, high stocking density; SEM, standard error of the means; INT, interaction.

## Data Availability

For 16S gene sequencing, raw sequence files (.fastq.gz) were submitted to the NCBI Sequence Read Archive (SRA) database and can be accessed here: https://www.ncbi.nlm.nih.gov/sra/PRJNA868544.

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
