# Peer review of "Evaluation of Liquid Organic Acids on the Performance, Chyme pH, Nutrient Utilization, and Gut Microbiota in Broilers under High Stocking Density"

_animals, 2023, doi:10.3390/ani13020257_

Round 1

Reviewer 1 Report

I have reviewed the manuscript entitled "Evaluation of Liquid Organic Acids on the Performance, Chyme pH, Nutrient Utilization, and Gut Microbiota in Broilers under High Stocking Density" (identified as animals-2046628). In this study, the authors investigated the effect of supplemental organic acids on high stocking density stress in grower broilers. An adequate amount of information has been provided in the introduction, and the researchers have clearly outlined their objectives. In other sections, there are, however, a few points that need to be improved.

Majorcomments:

Because of the word count limitation, certain words were omitted from the simple summary. For this reason, I believe that the objectives of the study should be stated in one sentence following two different sentences describing the reasons for conducting the study.

Results can be presented in a more striking manner.

Minor comments:

L43: An alphabetical sort of keywords is recommended.

L54: A dot should be added to the end of the sentence. Please use “[5,6].” Ä°nstead of “[5,6]”.

L61-63: A conjunction is missed in this sentence, in which two different emphases are placed.

L90, 102, 107, 272, 361, 363, 367, 369, 370, 373, 375, 424, 433, 486, 493, 494, 509, 515: Please use “HSD” instead of “high stocking density”.

L106, 107, 122, 123, 329, 330, 335, 336: Please use “organic acid added water” instead of “organic acids water”.

L127: It is helpful to specify the size of the cage used in the study.

L133: “ad libitum” should be italicized.

L128, 129, 144, 147, 149, 162, 170: Please give a space between “…” and “°C”.

L147: The method used to obtain the serum (rpm, time) should be described.

L162: Please use “3000 rpm/min” instead of “3000 r/min”.

L171:  Storing conditions should be specified.

L208-213: The information between L208-213 should be moved to the statistical analysis section.

L222: Please use “0.05 < p < 0.10” instead of “0.05 < p < 0.1”.

L227-229: Please remove one of the “in grower phase”.

L232, 247, 265, 281, 295, 305,  According to section 2.1, the starter phase consists of two groups of ten repetitions. The footnotes you have provided below the tables indicate that you have given these data as the average of six repetitions. This means that the footnote should be written more clearly.

L238, 239: In L222, you specified the significance level as p < 0.05, whereas here it is p ≤ 0.05. This is an inconsistency that should be corrected.

L263: It is recommended to leave a space between the table and the paragraph

L274-276: It is recommended that this section be rewritten. The use of "while" is problematic

L278: The table should start on a new page.

L294, 304: Due to the lack of values in Tables 5, Table 6, TableS1, S2 and S3; CON, OA, and Starter phase should be removed.

L298: There must be a space between L298 and L299.

L341-354: A more effective writing style is needed for this section.

L404: Please use “OAs” instead of “organic acids”.

L397-398, 402: Please use “TG” instead of “triglyceride”.

L398: Please use “LDLC” and “CON” instead of “low-density lipoprotein cholesterol” and “control”.

L477: “decreased Escherichia coli” instead of “decreased E.coli”.

L481: “Bacteroidetes, Firmicutes and Firmicutes/Bacteroidetes” instead of “bacteroidetes, firmicutes and firmicutes/bacteroidetes”.

L483: “Proteobacteria, Verrucomicrobia and Cyanobacteria” instead of “proteobacteria, verrucomicrobia and cyanobacteria”.

L491: “Alistipes” instead of “Alistipes”.

L500: “Christensenellaceae” instead of “Christensenellace-ae”.

Best regards.

Author Response

Thanks a lot for your comments. Please see the attachment. 

Reviewer 2 Report

Comments to authors:

The study has merit in its design and analyses and provides evidence that organic acids (OAs) can be used as an oral supplement to partially alleviate negative impacts of high stocking density in broilers. The study adds to the existing body of literature on the beneficial effects of OA supplementation on broilers health and performance. However, the following comments need to be addressed for better readability.

Overall comment:

In the Tables, consider adding the SEM values for each treatment group alongside their means instead of pooling them.

Minor comments:

Ln 13: replace reducted with reduce

Ln 16: what do you mean by “not well”? fully completed?

Ln 17: efficacy of OAs on What?

Ln 18: replace occurred with occurs and led with lead

Ln 20: delete could replace decrease with decreased

Ln 13: replace former with upper

Ln 21: increased the activity of digestive enzymes

Ln 23: delete the before minor

Ln 26: This study

Ln 32: insert and between duodenum and increased

Ln 54: Please replace what’s more with a more formal word

Ln 63: end the sentence with a period after residues and start a new sentence thereafter.

Ln 65: replace reducted with reduce

Ln 65: replace will be with can be

Ln 75: replace as kind of with similar to

Ln 85: replace supplemental with with supplementation

Ln 97: replace per with each

Ln 65: insert when the pH in between water and reached

Ln 113: Do you mean: OAs were not supplemented in water on the day of immunization in this study?

Ln 128: Please be specific with day/week rather than saying initial temperature.

Ln 138: redundant words between during

Ln 140: delete was

Ln 149: describe acronym VFA at its first use.

Ln 182: replace contained with containing

Ln 218: delete was

In Table 1: the unit of measurement is missing in 42d BW of grower phase

Ln 218: delete any

Ln 259: delete was

Ln 274: Do not start a sentence with while. If you do, add another statement without a period.

Ln 361: replace have with has

Ln 362: add and before gut microbiota

Ln 368: FCR in OA group tended to decrease by 2.13%

Ln 374: replace exist with has exert

Ln 378: replace prove with have any

Ln 380: Insert inclusion of  

Ln 382: what do you mean by normal station?

Ln 385: replace supplemental with supplemented

Ln 390: replace not affected with did not affect

Ln 411: replace can decrease with decreased

Ln 416: Insert due to the fact that

Ln 422: replace active with activate

Ln 428: replace in with by

Ln 448: replace would with can

Ln 448: what do you mean by formatting?

 Ln 452: what kind of abilities?

Ln 453: consider replacing effects with roles

Ln 464: consider replacing ability with functions

Ln 475: replace supplemented with supplementation of

Ln 478: illustrated that

Ln 479: what do you mean by resisted?

Ln 479: such as

Ln 479: what do you mean by resisted?

Ln 487: rephrase this sentence

Ln 501: Please replace what’s more with a more formal word

Ln 507: replace supplemental with supplementation

Ln 508: delete value

Ln 511: replace supplemental with supplementation

Author Response

(The authors gave the same response as above.)

Reviewer 3 Report

This is an interesting topic, however this paper will require a significant amount of editing. 

Lines 87 and 116: this describes the arrangement of treatments, not the experimental design.  The experimental design of the study should be stated.   

Author Response

(The authors gave the same response as above.)

Round 2

Reviewer 3 Report

Lines 105-107 do not state the experimental design of the study.  What is stated is a description of the treatments, not the experimental design.  Please clarify the experimental design.

Author Response

Thanks a lot. Please see the attachment. 
